# Few-Shot Fine-Grained Image Classification via GNN

**DOI:** 10.3390/s22197640

**Published:** 2022-10-09

**Authors:** Xiangyu Zhou, Yuhui Zhang, Qianru Wei

**Affiliations:** School of Software, Northwestern Polytechnical University, Xi’an 710129, China

**Keywords:** deep learning, few-shot learning (FSL), fine-grained image classification, graph neural network (GNN)

## Abstract

Traditional deep learning methods such as convolutional neural networks (CNN) have a high requirement for the number of labeled samples. In some cases, the cost of obtaining labeled samples is too high to obtain enough samples. To solve this problem, few-shot learning (FSL) is used. Currently, typical FSL methods work well on coarse-grained image data, but not as well on fine-grained image classification work, as they cannot properly assess the in-class similarity and inter-class difference of fine-grained images. In this work, an FSL framework based on graph neural network (GNN) is proposed for fine-grained image classification. Particularly, we use the information transmission of GNN to represent subtle differences between different images. Moreover, feature extraction is optimized by the method of meta-learning to improve the classification. The experiments on three datasets (CIFAR-100, CUB, and DOGS) have shown that the proposed method yields better performances. This indicates that the proposed method is a feasible solution for fine-grained image classification with FSL.

## 1. Introduction

FSL could be considered as deep learning with few labeled samples. Traditional deep neural networks (DNN) usually require a large number of high-quality training samples without bias to avoid overfitting [1,2,3,4,5,6,7,8]. However, due to a number of factors such as privacy, security, or the high cost of labeling data, many real-world application scenarios cannot obtain enough labeled training samples. Therefore, it is crucial to investigate how a deep learning system can efficiently learn and generalize its cognitive capabilities from a small number of samples. This is especially true for fine-grained image classification where data acquisition is difficult and labeling is expensive.

Fine-grained image classification involves distinguishing basic categories and then producing fine subcategories, such as bird species, car models, dog breeds, etc. Currently, there is a wide range of business needs and application scenarios in industry and in real life. As shown in Figure 1, fine-grained images have more similarity in appearance and features than coarse-grained images. In addition, various influences such as pose, perspective, illumination, occlusion, and background interference can result in large inter-class differences and small intra-class differences in the data, making classification challenging. The traditional method of classification by assessing interclass and intraclass distances in Euclidean space does not seem to work well for fine-grained FSL. Using GNN to assess the similarity out of Euclidean space can solve this problem.

GNN is a neural network model for representation learning of data suitable for graphical representation in non-Euclidean space. Based on an information diffusion mechanism, GNN updates node states by exchanging neighborhood information recurrently until a stable equilibrium is reached [9]. The application of GNN in the field of few-shot classification can also be regarded as a metric learning method, and the relationships between samples can be obtained by GNN. Due to its efficient performance and interpretability, classification algorithms based on GNN have gradually gained acceptance.

In this paper, we propose a GNN-based fine-grained (GFF) image classification framework that implicitly models the in-class similarity and inter-class difference based on node tags. In this graph model, the features (output of the embedding model) and label information are used as input nodes, and the query sets without labels are classified by iteratively updating the distance between each node. In the experiments, we found that the quality of the feature extraction model has a direct impact on the final classification results, and that a poor embedding model is likely to cause the whole network to fall into the local optimum. With this in mind, we optimize the embedding model to maximize the distance between samples of different categories during feature extraction, which effectively improves the classification effect. In summary, our contributions are in the following aspects:We are the first to explore and use the information transfer of GNN for few-shot fine-grained image classification. The proposed method can better distinguish the nuances between different categories compared to other classification models.We optimize the embedding model. The initial embedding model is learned by the meta-learning method, which can make the feature extractor effective for both unknown and known test classes, and prevent it from falling into a local optimum.

We conduct extensive experiments and demonstrate the effectiveness of the proposed method.

## 2. Related Work

### 2.1. Few-Shot Learning

FSL is typically required to construct a new framework of a neural network using prior knowledge, which can be classified in three ways: (1) constructing external memory and introducing prior knowledge into the memory, (2) introducing prior knowledge into the initialization parameters of the model, and (3) using the training data as prior knowledge.

Models such as Convolutional LSTM [10], GRU [11], a variant of LSTM, and Neural Turing Machine [12], which improves the read and write operation of LSTM, are applied in the field of small-sample image classification with the idea of introducing an external memory and modifying the external memory in terms of training sets during training and use the external memory as prior knowledge during testing. The introduction of prior knowledge in the initialization parameters, i.e., the meta-learning method, aims to give the model a kind of learning capability that allows it to automatically learn some meta-knowledge. Meta-knowledge refers to the knowledge that can be learned outside the model training process, such as the hyperparameters of the model, the structure and optimizer of the neural network, etc. Typical networks include MAML (Model-Agnostic Meta-Learning), proposed by Finn et al. [13] in 2017, and Meta-SGD [14], an SGD-like, easily trainable meta-learner, which was developed based on MAML so that the model can learn the direction of optimization and learning rate. In addition, there is also the Reptile, a scalable meta-learning algorithm, proposed by Alex et al. [15] in 2018, which avoids the two derivatives of MAML and greatly reduces the calculation.

Currently, the common few-shot classification algorithm uses training data as prior knowledge, including a fine-tuning-based model and a metric-based model. The idea of the former is to pre-train the model using large-scale data and fine-tune parameters using a target few-shot dataset, and the idea of the latter is to extract the features of the samples through an embedding network for distance analysis to obtain a network that can perform the relationship discrimination and matching of the different categories. Typical networks for metric learning include the Prototypical Network, proposed by Snell [16] et al., whose idea is to extract prototypical features from samples of the same category and assess the sample category by comparing the Euclidean distance between the query sample and the prototypical features; the Matching Network [17], which introduces an attention mechanism; and Relation Network [18], which assesses the relations using adaptive nonlinear classifiers.

### 2.2. Fine-Grained Image Classification

The core of fine-grained image classification methods is to focus on the details of the image to extract discriminative features for classification. Currently, popular methods can be summarized into three categories: (1) parts localization-based methods, (2) end-to-end feature learning-based methods, and (3) feature enhancement-based methods.

Early methods usually introduce auxiliary annotation information to locate the key components of the target. For example, the Part R-CNN proposed by Zhang et al. [19] and the Deep LAC model proposed by Lin et al. [20] generate multiple regions, then localize the semantic components of the target using the annotation information, and finally extract the features of the local regions for classification. Since the above methods need to introduce additional annotation information, they failed to become mainstream methods for fine-grained image classification. Researchers focus more on weakly supervised methods, i.e., end-to-end methods based on discriminative feature learning, which directly extract more discriminative features for classification by developing a powerful deep model or a new loss function. The core of the RA-CNN model proposed by Fu et al. [21] and the MAMC model combined with metric learning [22] is to gradually focus on key regions through the attention mechanism, and determine the final result based on the classification of the features of these regions. The main technical route of the Bilinear CNNs, proposed by Lin et al. [23], and the Compact Bilinear model, proposed by Yang et al. [24], is to use two branches to extract image features, then fuse the top-level features, and finally obtain a high-order feature that fuses all channel information for classification.

In recent years, some methods have combined the idea of local feature location and discriminative feature learning to obtain strong discriminative features from the perspective of feature enhancement. To make the model focus on the refined regions rather than the overall components, Chen et al. proposed a special learning model, DCL [25]. In this method, the image scrambled in a certain local region and the original image are input to the model together to destroy the overall structure of the image. The model is then forced to learn the local details of the image, so that the features extracted by the model can be enhanced. The WS-DAN model, proposed by Hu et al. [26], introduces a data augmentation framework that first generates an attention map through weakly supervised learning to represent multiple discriminative parts of the target, and then uses two data augmentation methods to strengthen the discrimination of features.

### 2.3. FSL for Fine-Grained Classification

With the gradual deepening of research on few-shot learning and fine-grained image classification, researchers begin to consider how to solve the problem of few-shot learning for fine-grained image classification. Wei et al. proposed the problem of Few-Shot Fine-Grained image classification (FSFG) in [27], and proposed a matching network PCM for slicing bilinear features. The method obtains the local features of the target by decomposing bilinear features, and adopts task-oriented learning in the learning process to adapt to the FSL problem. Subsequently, Li et al. proposed two few-shot fine-grained classification models, CovaMNet and DN4, to measure the relationship between images by the relationships between local features [28,29]. Although the above methods focus on the importance of local regions, they ignore the influence of irrelevant local regions. In this context, Hou et al. proposed a cross-attention network, CAN [30], to localize relevant regions, which uses the attention mechanism to reweight local features.

Similarly, Zhang et al. proposed the DeepEMD model with a new weighted distance formula [31] and reached the leading level. They considered the few-shot classification problem as an optimal matching problem, divided the image into multiple patches, and then introduced the distance metric EMD to calculate the best matching cost between each patch of the image to represent the similarity between them, to realize the mining of key regions.

### 2.4. GNN-Based Methods in FSL

Recently, GNN has been widely used in the field of few-shot learning. Specifically, Garcia et al. first utilized GNN to solve few-shot learning problems, where the embedding model and GNN model were trained end-to-end as one [32]. Liu et al. proposed a transductive propagation network (TPN). The TPN utilizes the entire query set for transductive inference to further exploit intra-class similarity and inter-class dissimilarity [33]. Kim et al. proposed an edge-labeling graph neural network [34]. Gidaris et al. reconstructed the classification weights using a denoising autoencoder network on the GNN-based few-shot model [35]. To explicitly model the distribution-level relation, Yang et al. proposed the distribution propagation graph network (DPGN) [36]. The GNN-based model is significant to be explored widely because of its interpretability and acceptable performance.

## 3. Materials and Methods

### 3.1. Data Organization and Definition of Task

The dataset consists of the training set $D_{\mathrm{tr}}$ (including $T_{\mathrm{tr}}$ and $T_{\mathrm{te}}$) and the test set $D_{\mathrm{te}}$ (including $T_{\mathrm{tr}}$ and $T_{\mathrm{te}}$). These two sets are disjoint, i.e., $D_{\mathrm{tr}}\cap D_{\mathrm{te}}=\emptyset$, to ensure that the trained model is also suitable for new tasks. 

In the N-way K-shot problem, the number of categories is N, the number of support samples of each category is K, and the training data is sampled as follows: N classes are randomly selected in $D_{\mathrm{tr}}$, and K samples are randomly selected for each of the N classes to construct $T_{\mathrm{tr}}$ (i.e., there are N×K support samples in $T_{\mathrm{tr}}$). A sample from one of these classes is randomly selected as a query sample to construct $T_{\mathrm{te}}$, and $T_{\mathrm{tr}}\cap T_{\mathrm{te}}=\emptyset$. During the training of each task, all samples in $T_{\mathrm{tr}}$ and $T_{\mathrm{te}}$ of $D_{\mathrm{tr}}$ are input into the model. The label of the query sample is used as the ground truth for prediction.

For testing, $T_{\mathrm{tr}}$ and $T_{\mathrm{te}}$ are constructed on $D_{\mathrm{te}}$ in the same way as for training. The training and testing tasks share the same N-way K-shot problem, i.e., there are also N×K labeled samples and an unknown sample that are input into the model when testing. The output category is the prediction label of test sample (i.e., $T_{\mathrm{te}}$).

### 3.2. Model Design

The whole model of this algorithm is shown in Figure 2. First, the embedding model is pre-trained by meta-learning, the trained model is used as the initial embedding model for formal training, and then the embedding model φ(x) and GNN model are trained end-to-end.

The embedding model (feature extraction model) is composed of four convolutional layers to process each 32 × 32 image with three channels into a 64-dimensional feature vector, which is input as a node into the graph in the following work. The process is shown in Figure 3. Our work aims to address the few-shot problem. In supervised learning, the insufficient training examples can lead to overfitting. Compared with the limited knowledge from few examples, the deep models with a large number of learning parameters are too complicated [13,14]. Therefore, we chose a lightweight embedding model with fewer parameters and low depth.

The GNN model (classification model) was developed to obtain the distribution of features for classification. There are N×K+1 nodes, representing the embedding vectors of all samples in the task T. There are edges between every two nodes in the graph. The weight of each edge represents the relationship between two images (which can be understood as distance or similarity).

As shown in Figure 4, an iteration of GNN contains two modules: Adjacency Block and Convolution Block. After the Adjacency Block is executed once, the connection between nodes is changed once. After the Convolution Block is executed once, the features of the nodes are changed once.

### 3.3. Pre-Training Feature Extractor

Before formal training, the feature extractor is pre-trained. The feature extraction model is optimized by meta-learning. Then, it is connected with a full-connection layer as a classifier model f to start training.

Based on supervised learning, the model f is trained based on the empirical loss $L_{\mathrm{tr}}$ by SGD, expressed in the following equation:(1)$$\theta^{\prime }=\theta -\beta \nabla L_{\mathrm{tr}}$$
where β is the learning rate of the base-learner. Unlike in [13], it is learnable and not manually defined. The meta-learner aims to learn an acceptable θ and β that can guide the base-learner to obtain the temporary parameters $\;\theta^{\prime }$ for the current task. The setting of learnable β can make it more automatic to update the parameters θ to avoid overfitting or underfitting. 

The training loss of the task, $L_{\mathrm{tr}},$ is computed as follows:(2)$$L_{\mathrm{tr}}=\frac{1}{\left|T_{\mathrm{tr}}\right|}\underset{\left(x,y\right)\in T_{\mathrm{tr}}}{\sum }l\left(f_{\theta }\left(x\right),y\right)$$
where l represents the cross-entropy loss, x is the input image with the label y, and $\left|T_{\mathrm{tr}}\right|$ is the number of training samples for this task.

Mathematically, the objective of the proposed meta-learning model can be formulated as follows:(3)$$\underset{\theta ,\beta }{\min}L_{\mathrm{te}}=\underset{\theta ,\beta }{\min}\frac{1}{\left|T_{\mathrm{te}}\right|}\underset{\left(x,y\right)\in T_{\mathrm{te}}}{\sum }l\left(f_{\;\theta^{\prime }}\left(x\right),y\right)$$
where $L_{\mathrm{te}}$ is the test loss of each task, and $\left|T_{\mathrm{te}}\right|$ is the number of test samples for the current task.

With the above analysis, we can summarize the training process of the meta-learning model as follows:(4)$$\left(\theta ,\beta \right)\leftarrow \left(\theta ,\beta \right)-\alpha \underset{T_{i}\in p\left(T\right)}{\sum }\nabla L_{\mathrm{te}}$$
where α is the manually determined learning rate for training the meta-learner, and p(T) is the distribution of a set of tasks in a meta-training epoch.

The details of the pre-training are shown in Algorithm 1.
**Algorithm 1.** Pre-training feature extractor**Input:** Meta-learner learning rate α, the pre-training model f, few-shot task distribution p(T), the batch size I**Output:** good initialization parameters $\theta^{*}$, learnable base-learner learning rate β 1: Randomly initialize θ, β 2: **for** i=1; i≤I; i=i+1 **do**3:  Sample n tasks from p(T);4:  **for** j=1; j≤n; j=j+1 **do**5:   $L_{T_{\mathrm{tr}}^{j}}=\frac{1}{\left|T_{\mathrm{tr}}^{j}\right|}\underset{\left(x,y\right)\in T_{\mathrm{tr}}^{j}}{\sum }l\left(f_{\theta }\left(x\right),y\right)$;6:   Backward $L_{T_{\mathrm{tr}}^{j}}$ as $\nabla L_{\mathrm{tr}}$;7:   $\theta_{j}=\theta -\beta \nabla L_{\mathrm{tr}}$;8:   $L_{T_{\mathrm{te}}^{j}}=\frac{1}{\left|T_{\mathrm{te}}^{j}\right|}\underset{\left(x,y\right)\in T_{\mathrm{te}}^{j}}{\sum }l\left(f_{\theta_{j}}\left(x\right),y\right)$;10:  **end for**11:  Backward $\overset{n}{\underset{j=1}{\sum }}L_{T_{\mathrm{te}}^{j}}$ as $\nabla L_{\mathrm{te}}$;12:  Update $\left(\theta ,\beta \right)\leftarrow \left(\theta ,\beta \right)-\alpha \nabla L_{\mathrm{te}}$;13: **end for**

### 3.4. Formal Training

After pre-training, the embedding model is created with initial parameters. In the formal training stage, it is merged with the GNN classification model and a fully connected layer to form the whole model g for end-to-end training.

In each training round, the support samples and query samples are input to the embedding model. Then, the feature is generated and concatenated with the labels to form a node set. The initial node set expression is shown in (5):(5)$$Y={\left\{\left(\emptyset \left(x_{i}\right),h\left(x_{i}\right)\right)\right\}}_{i\in \left[1,N\times K+1\right]}$$
where $\emptyset \left(x_{i}\right)$ represents the feature vector of image $x_{i}$ obtained by the feature extraction model, and $h\left(x_{i}\right)$ represents the one-hot encoding of the category of image $x_{i}$. Then, the nodes are input to the GNN model for classification training, including the creation of the adjacency matrix (Adjacency Block) and the modification of features (Convolution Block).

Adjacency Block: The relationship between every two nodes is expressed as follows:(6)R(i,j)=MLP(abs(i,j))
where MLP is a multi-layer neural network whose input is the Euclidian distance between two nodes. After obtaining all R(i,j), adjacency matrix A can be constructed.Convolution Block: The new nodes are obtained by a graph convolution neural network, which is formulated as follows:(7)$$X^{\left(k+1\right)}=\rho \left(\sum \theta^{\left(k\right)}A^{\left(k\right)}X^{\left(k\right)}\right)$$
where ρ is a nonlinear activation function, θ is a learned parameter, and X is the matrix of features.

After several iterations of learning the feature distribution and learning potential features, an adjacency matrix can be created to represent the final distribution.

Finally, a fully connected layer is applied to output the possibility of distributed categories, which is used to calculate the cross-entropy loss: (8)$$L=\overset{N}{\underset{i=1}{\sum }}y_{\mathrm{que}}\log\left(y_{\mathrm{que}}^{*}\right)$$
where N is the number of categories, $y_{\mathrm{que}}$ is the label of the query sample $x_{\mathrm{que}}$ in $T_{\mathrm{te}}$, and $y_{\mathrm{que}}^{*}$ represents the categories of possibility distribution of $x_{\mathrm{que}}$.

We update the parameters of the whole model by back-propagation:(9)$$\;\theta^{\prime }=\theta -\eta \nabla L\left(\theta \right)$$
where θ is the parameter set before update, η is the learning rate, and ∇L(θ) represents the partial derivative of the loss with respect to θ.

The details of formal training are shown in Algorithm 2.
**Algorithm****2****.** Formal training**Input:** Meta-learner learning rate η, the whole model g (including embedding model ∅ with $\theta^{*}$), few-shot task distribution p(T), the number of batches N, the batch size I, the number of iterations n**Output:** good parameters $\;\hat{\theta }$ 1: Randomly initialize other parameters (θ refers to all the parameters of g); 2: **for** i=1; i≤N; i=i+1 **do**3:  Sample I tasks from p(T);4:  for j=1; j≤I; j=j+1 do 5:   Get nodes $Y={\left\{\left(\emptyset \left(x_{i}\right),h\left(x_{i}\right)\right)\right\}}_{i\in \left[1,N\times K+1\right]}$; 6:   Get the initial matrix of features $X^{\left(0\right)}$;7:   **for** k=1; k≤n; k=k+1 **do**8:     R(i,j)=MLP(abs(i,j));9:     $X^{\left(k\right)}=\rho \left(\sum \theta^{\left(k-1\right)}A^{\left(k-1\right)}X^{\left(k-1\right)}\right)$;10:   **end for**11:   $L=\overset{N}{\underset{i=1}{\sum }}y_{\mathrm{que}}\log\left(y_{\mathrm{que}}^{*}\right)$;12:  **end for**13:  Backward $\overset{I}{\underset{j=1}{\sum }}L$ as ∇L;14:  Update θ←θ−η∇L(θ);15: **end for**

## 4. Experiment and Discussion

### 4.1. Dataset

The experiments are mainly conducted on a public dataset, CIFAR-100, with 100 categories of 600 color images of size 32×32 each, collected by Krizhevsky et al. [37]. As shown in Table 1, each image has a class label and a super class label for fine- and coarse-grained classification, respectively. We use the class label as the experiment label.

We import CIFAR-100 directly from the torchvision.datasets library and normalize the images into a dataset. During training, we set the random seed to 1. For the N-way task, N categories in 100 categories were randomly selected as a test set $D_{\mathrm{te}}$, and the remaining data were used as the training set $D_{\mathrm{tr}}$. In the pre-training stage, $D_{\mathrm{te}}$ and $D_{\mathrm{tr}}$ are divided in the same way as in the formal training stage to ensure the rationality of the experiment, i.e., the categories included in $D_{\mathrm{te}}$ and $D_{\mathrm{tr}}$ remained unchanged throughout the experiment.

To prove the validity threats of the proposed technology, we also evaluated the proposed method on other two datasets: (1) Caltech-UCSD Birds-200-2011 (CUB) [38], which contains 11,788 images of 200 sub-categories of birds, and (2) Stanford Dogs (DOGS) [39], which consists of 20,580 images from 120 dog species.

### 4.2. Network Structure Setting

The embedding architecture consists of four convolutional layers resulting in a 64-dimensional feature embedding [32]:

{3 × 3-conv(32 filters), batch normal, relu, max pool(2, 2)}; 

{3 × 3-conv(64 filters), batch normal, relu, max pool(2, 2)}; 

{3 × 3-conv(128 filters), batch normal, relu, max pool(2, 2)}; and

{3 × 3-conv(64 filters), batch normal, relu}.

This simple architecture is suitable for fast prototyping. When pretraining the embedding model, a softmax layer is added after it to output the probability distribution of various classes.

As shown in Figure 5, the GNN model consists of three Adjacency Blocks and three Convolution Blocks that are stacked alternately. The last block is for classification. The Convolution Block consists of a matrix multiplication operation, a linear layer, a batch-normalization layer, a ReLu layer, and a concatenation operation. The Adjacency Block contains an operation to calculate Euclidean distance and three convolutional layers: 

{1 × 1-conv(64 filters), batch normal, relu}; 

{1 × 1-conv(32 filters), batch normal, relu}; and

{1 × 1-conv(1 filter)}.

The input are N×K+1 features, each of which is (64 + *N*)-dimensional. The output is a probability distribution of categories.

### 4.3. Experimental Settings

Pre-training is carried out in the way of supervised learning. The Adam optimizer is used with a learning rate of 0.01. We perform meta-training of the models for 50,000 episodes, each of which contains 32 internal tasks (i.e., the batch of tasks is 32). For the N-way K-shot task, N categories are randomly selected in $D_{\mathrm{tr}}$ during training. K samples are randomly selected in each category to form the training subset, $T_{\mathrm{tr}}$, which contains N×K images in formula (2). The test subset $T_{\mathrm{te}}$ in Formula (3) is created by selecting N categories from the other categories in $D_{\mathrm{tr}}$ and randomly selecting K samples in each category. We update the optimizer in each epoch, calculate the accuracy of the model in the validation set every 1000 epochs, and record the minimum loss. It should be noted that the samples selected in the pre-training stage are all from the training set $D_{\mathrm{tr}}$, which ensures that each query sample in the test set is evaluated independently.

In the pre-training process, the labels are set as follows: the selected categories in each task are randomly used from 0 to N-1 as label identifiers, representing N of all categories in CIFAR-100. In this way, the convenience of the experiment is realized and the diversity of tasks is ensured to improve the generalization effect of the model. For example, as shown in Figure 6, the two tasks could randomly select some categories that are the same, while the class rose is marked 0 in the Task 1 and 1 in the Task 2. In predictive distribution, this label is predicted in two tasks with different positions in the predictions of the classifier. This approach to labeling is also used for formal training and testing procedures.

In formal training, we set the batch size of the training stage to 16 based on experience and comparison of multiple experiments. We train the whole model for 20,000 episodes, and each episode contains 16 internal tasks. For each task, we construct $T_{\mathrm{tr}}$ in the same way as in pre-training, and randomly select a sample from one of the N classes included in $T_{\mathrm{tr}}$ as the query sample (i.e., $T_{\mathrm{te}}$). Each task has N×K+1 sample images as the input of the model. The loss is calculated based on the comparison between the output and the actual label number. The back-propagation is performed once in each round. The Adam optimizer with the initial learning rate of 0.01 is used. Then, the learning rate is reduced exponentially by:(10)$$\eta_{\tau }=\eta_{0}\times g^{\frac{\tau }{T}}$$
where $\eta_{0}$ is the initial learning rate, g =0.9 is the hyperparameter, τ is the current iteration number, and T =20,000 is the total number of training epochs.

In the testing process, K images are randomly extracted from each category of $D_{\mathrm{te}}$, containing only N target categories to construct $T_{\mathrm{tr}}$. One image is randomly selected from one of the categories as $T_{\mathrm{te}}$. We set the parameters of the whole model and feed these N×K+1 images into the network to obtain the classification result. To ensure the efficiency of the experiment, after every 1000 rounds of training, a round of testing on the $D_{\mathrm{te}}$ is conducted to calculate the loss and accuracy, and the lowest loss was recorded. The training was stopped when the lowest loss was not updated for 10 consecutive times.

All the experiments were carried out on a workstation with Intel Core i7-11700K CPU (eight cores, 3.80 GHz) and the NVIDIA GeForce RTX 3060 GPU (12 GB memory).

### 4.4. Ablation Study

We performed various experiments to evaluate the effectiveness of our algorithm. All the experiments were performed on the CIFAR-100 dataset, if not stated otherwise. The method of [32] was set as the baseline for ablation experiments.

We performed several ablation experiments to prove that the optimization of the feature extraction is useful for both unknown and known test classes, and can avoid falling into a local optimum. Taking the five-way one-shot as an example, if we directly connect the embedding model and the GNN model for end-to-end training, following the method of [32], the classification accuracy is about 36%, and the loss value was low enough. We suspect that the whole network may fall into a local optimum, and the optimized embedding model can solve this problem. To check this, we fix the embedding model and update only the GNN model. The accuracy is about 30%. However, if we fix the embedding model after optimization, the accuracy is up to 41%.

To further prove the importance of optimizing the embedding model, we compared the accuracy of GNN with or without pre-training on CIFAR100, CUB, and DOGS, as shown in Table 2.

In formal training, we try to optimize the network parameters only once for N classes randomly selected for each task according to the testing process of meta-learning [4], and then perform end-to-end training for the whole model after obtaining a specific embedding model. However, the effect is weak. The reason for the analysis is that the parameters of the embedding model change too much during the training of each task, which causes the failure of convergence of the whole model. We assume that this problem can be improved if the parameters of the specific embedding model are fixed to train the model, i.e., only the GNN classification model is trained. Although the effect is improved, the accuracy of five-way one-shot is still only about 32%. The reason is that the classification criteria of feature extraction and the classifier are different in this way, and the criteria applicable to the embedding model directly used for the classifier cannot achieve the desired effect when the embedding model is required as the feature extractor. However, the two parts must be consistently matched to support each other in FSL. Thus, it can be confirmed that the classification accuracy depends on the effectiveness of the learned embedding network and is therefore limited by the insufficient differential representations of the embedding network.

We also try to change the pre-training section for a full classification of all categories. The accuracy is improved compared to that of the GNN classification model without pre-training, as shown in Table 3. However, it does not reach the effect of the proposed method. We suspect that the full classification results in the embedding model overfitting to all categories in training data. In contrast, the feature extractor can be better applied to unknown test categories by learning the initial embedding model through meta-learning. For the five-way tasks, the classification accuracy is 12.5%, 5.2%, and 3.2% higher than that of the original GNN-based classification model for 1-shot, 5-shot, and 10-shot, respectively, and the effects are significantly improved. This approach is more in line with the idea of formal training and effectively helps the formal training to avoid falling into a local optima.

In general supervised learning, when the number of training samples is insufficient, the supervised learning model could easily have an overfitting problem. To solve this problem, some data enhancement operations such as random cropping, mirroring, and resizing are usually performed to increase the number of training images. Similarly, the proposed task-oriented model with insufficient training tasks will also suffer from the overfitting problem. We also tried to perform data enhancement in our experiment to achieve a better performance, but the experimental results were not satisfactory. The method is referred to as “GNN+Pre-training (meta-learning) + Enhancement” in Table 3. As shown in Figure 7, after adding data augmentation, the model converges slightly faster than before, but the accuracy is not improved. The reason is analyzed as follows: Unlike general supervised learning, where the number of training samples plays an important role, for task-oriented learning, the more training samples, the better the performance. However, for the few-shot problem, it is difficult to increase the number of training categories, and task expansion may not be effective.

### 4.5. Performance and Analysis

Several representatives of general neural networks, meta-learning-based and metric-based classification methods, are selected for comparison on CIFAR-100. For the ordinary deep-learning-based methods, there are k-means [40] and DC [41]. For the meta-learning-based methods, we select MAML [13] and MetaCurvature [42], which is an enhanced version of MAML. For the metric-based methods, ProtoNet [16], MatchNet [17], and TADAM [43] are selected. In addition, DeepEMD [31], IDeMe-Net [44], MixtFSL [45], IER [46], MCRNet [47], and MTL [48] are also used as state-of-the-art methods.

We compare GFF to other methods, splitting the data into the test set and the training set in the same way. For the five-way one-shot and five-shot experiments, we repeatedly take 500 testing episodes on CIFAR-100 and record their average accuracy. The performance comparison is shown in Table 4. As shown in the table, our results significantly outperform the state-of-the-art models, e.g., one-shot (3.8%) and five-shot (6%). Compared to the original GNN-based model, other methods perform better for one-shot, but not for five-shot. Therefore, we infer that GNN might be more useful in the representation of the distance between samples in the case of multiple shots. It is obvious that our method performs significantly better than other baselines, due to the optimized feature extractor.

Earlier related few-shot works such as ProtoNet use fixed distance measures such as Euclidean or cosine distance for classification [16]. All learning in these metric learning-based studies takes place in feature embeddings. Compared to the previous fixed metrics or fixed features and shallow learning metrics, RelationNet [18], which can be viewed as learning of both a deep embedding and a deep nonlinear metric (similarity function), can achieve better classification performance. It learns a suitable metric in a data-driven manner by using a flexible functional approximator to learn the similarity between classes, without having to manually select the correct metric (Euclidean, cosine, etc.). Based on the idea of RelationNet, the proposed method changes similarity function from deep nonlinear metric to GNN, and uses its information transfer property to measure the relationships between inter-classes and intra-classes more flexibly and effectively for fine-grained FSL. Thus, better results can be obtained.

To demonstrate the generality of the proposed method in the fine-grained FSL domain, we compare the method with state-of-the-art methods on CUB and DOGS. As shown in Table 5, GFF achieves leading results in most cases.

The running times of the testing process on CIFAR-100 for the partial compared methods at the same batch-size setting are shown in Figure 8, and analyzed as follows: The metric-based methods (i.e., ProtoNet, MatchNet, and TADAM) run longer than the ordinary deep-learning-based methods (K-Means, DC). This is because the class labels are predicted by metric-based methods through calculating the distance or similarity scores between the sample features and the prototypes of each class, which increases the computational cost. Moreover, since the prototype of each class is computed from the support set in the target domain, metric-based methods require more running time as the number of samples in the training set increases. MAML, which is based on meta-learning, predicts the category of an image by the fully connected layer. Therefore, its running time is less than that of the metric learning methods. The running time of the MetaCurvature model increases sharply due to the large number of parameters, which is the longest among all the compared models.

In the proposed method, the model complexity of the computation part of the distance is slightly higher than that in other metric learning methods, as is the running time. However, a slight increase in model complexity and running time is acceptable considering the significant performance gain of the model.

## 5. Conclusions and Future Work

In this paper, we propose the GFF model to solve few-shot fine-grained image classification. By obtaining the relational structure of each sample, the method can calculate the similarity between images more reasonably than other methods that only learn the sample representation. Particularly, we designed an optimizer of the embedding model based on meta-learning to obtain a better initial model, which effectively improves the classification performance and allows the whole network to converge rapidly. This is a unique attempt. Extensive experiments have shown that GFF achieved the best performance of all compared transfer learning solutions on the CIFAR-100 dataset. However, the limitation of the proposed method is that the training time cost is too large.

From the perspective of pre-training, our experiments confirm that the optimization of the embedding model is of great importance to improve the performance of the whole algorithm. The next study will focus on improving the embedding model by combining the attention mechanism to obtain a more useful feature representation. In addition, the diversity of training categories for FSL is particularly important. For some FSL problems with few categories, how to increase the number of categories may become a future research direction. In this context, it may be possible to combine the ideas of a Generative Adversarial Network [51] to create new categories for training.

## Figures and Tables

**Figure 1 sensors-22-07640-f001:**
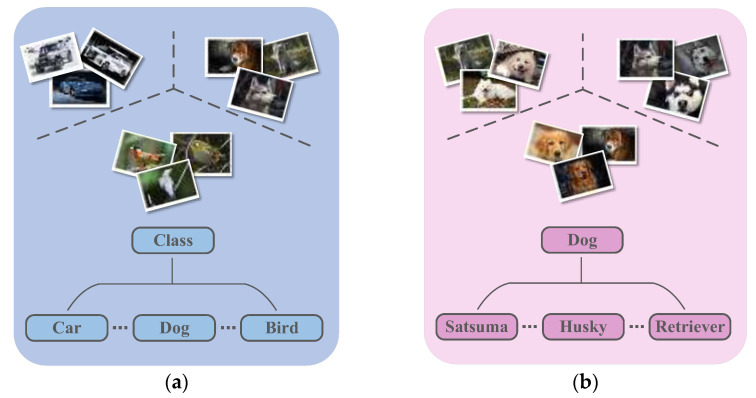
(**a**) Generic image recognition (coarse-grained). (**b**) Fine-grained image recognition.

**Figure 2 sensors-22-07640-f002:**
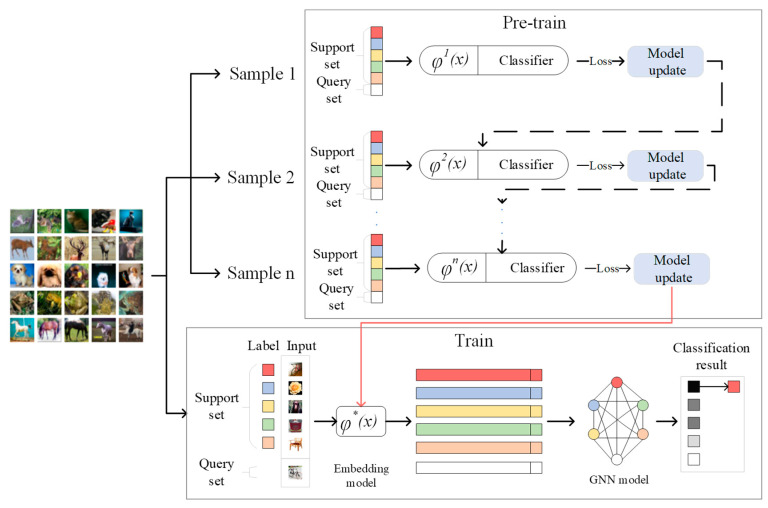
The whole process of the algorithm, including pre-training and formal training; φ(x) is the embedding model.

**Figure 3 sensors-22-07640-f003:**
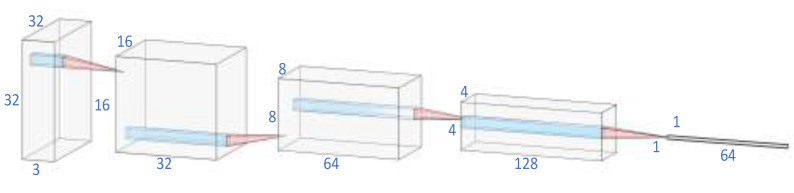
The process of feature extraction.

**Figure 4 sensors-22-07640-f004:**
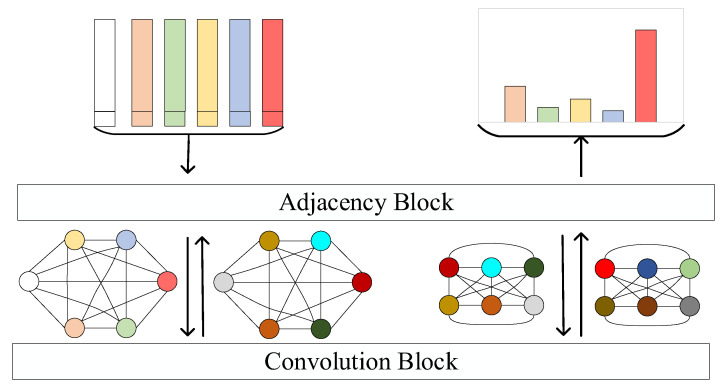
The training process of the graph neural network. Perform each Convolution Block after Adjacency Block.

**Figure 5 sensors-22-07640-f005:**
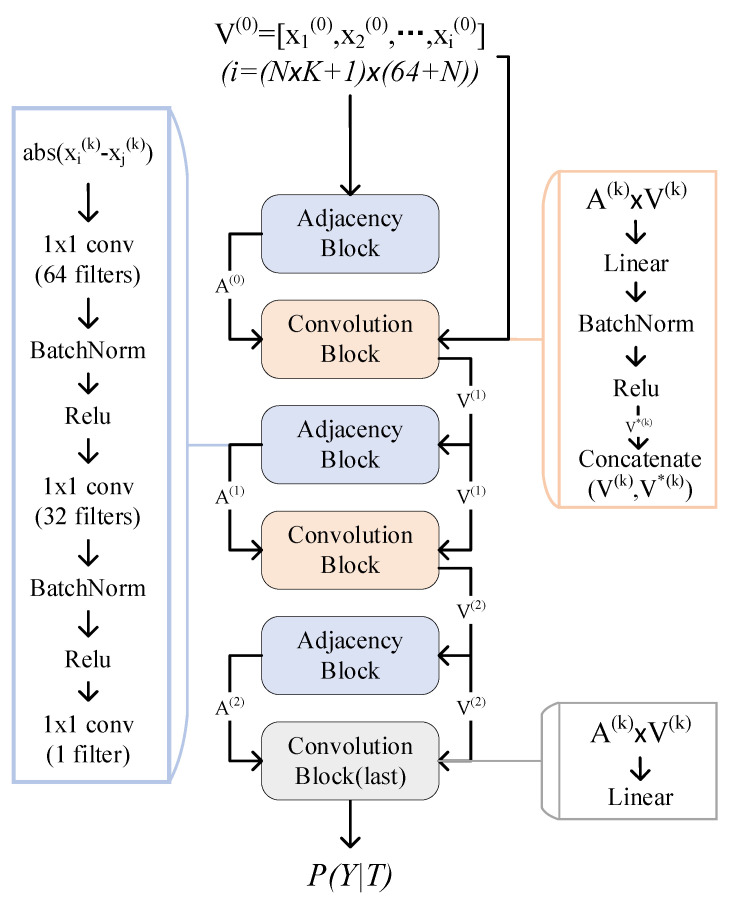
The architecture of the GNN model.

**Figure 6 sensors-22-07640-f006:**
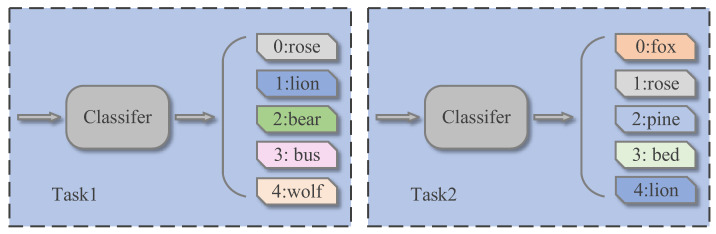
Two different types of tasks.

**Figure 7 sensors-22-07640-f007:**
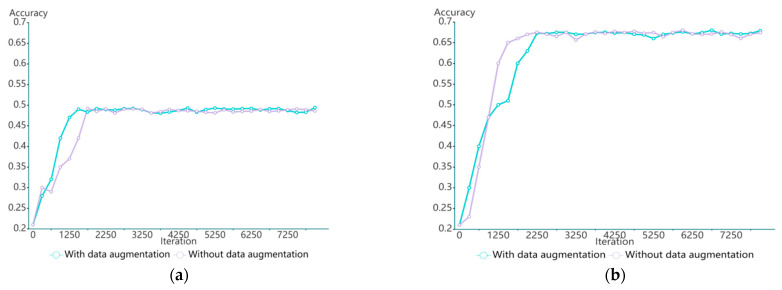
Classification accuracies with or without data enhancement in (**a**) the 5-way 1-shot case and (**b**) the 5-way 5-shot case.

**Figure 8 sensors-22-07640-f008:**
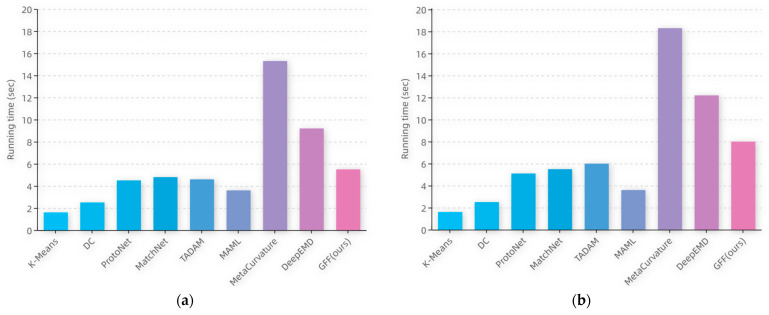
The running time of a single sample test phase for different methods in (**a**) the 5-way 1-shot case and (**b**) the 5-way 5-shot case.

**Table 1 sensors-22-07640-t001:** The labels of CIFAR-100.

Super Class	Classes
aquatic mammals	beaver, dolphin, otter, seal, whale
fish	aquarium fish, flatfish, ray, shark, trout
flowers	orchids, poppies, roses, sunflowers, tulips
food containers	bottles, bowls, cans, cups, plates
fruit and vegetables	apples, mushrooms, oranges, pears, sweet peppers
household electrical devices	clock, computer keyboard, lamp, telephone, television
household furniture	bed, chair, couch, table, wardrobe
insects	bee, beetle, butterfly, caterpillar, cockroach
large carnivores	bear, leopard, lion, tiger, wolf
large man-made outdoor things	bridge, castle, house, road, skyscraper
large natural outdoor scenes	cloud, forest, mountain, plain, sea
large omnivores and herbivores	camel, cattle, chimpanzee, elephant, kangaroo
medium-sized mammals	fox, porcupine, possum, raccoon, skunk
non-insect invertebrates	crab, lobster, snail, spider, worm
people	baby, boy, girl, man, woman
reptiles	crocodile, dinosaur, lizard, snake, turtle
small mammals	hamster, mouse, rabbit, shrew, squirrel
trees	maple, oak, palm, pine, willow
vehicles 1	bicycle, bus, motorcycle, pickup truck, train
vehicles 2	Lawn mower, rocket, streetcar, tank, tractor

**Table 2 sensors-22-07640-t002:** Classification accuracies (%) of GNN with or without pre-training.

	Pre-Training	Cifar100	CUB	DOGS
5-way 1-shot	-	36.7	51.8	47.0
	√	49.2 (↑ 12.5)	61.1 (↑ 9.3)	49.8 (↑ 2.8)
5-way 5-shot	-	62.3	73.7	63.3
	√	67.5 (↑ 5.2)	78.6 (↑ 4.9)	65.3 (↑ 2.0)

**Table 3 sensors-22-07640-t003:** Classification accuracies (%) with different pre-training methods in the 5-way K-shot cases.

Method	1-Shot	5-Shot	10-Shot
GNN (Raw, Baseline)	36.7	62.3	69.3
GNN + Pre-training (full classification)	47.3	63.8	68.9
GNN + Pre-training (meta-learning)	49.2	67.5	72.5
GNN + Pre-training (meta-learning) + Enhancement	49.5	67.6	72.3

**Table 4 sensors-22-07640-t004:** Classification accuracies (%) with different methods in the 5-way K-shot cases on CIFAR-100.

Method	1-Shot	5-Shot
K-Means [40]	38.5 ± 0.7	57.7 ± 0.8
DC [41]	42.0 ± 0.2	57.0 ± 0.2
MAML [13]	41.7 ± 0.6	56.2 ± 0.6
MetaCurvature [42]	41.1 ± 0.7	55.5 ± 0.8
ProtoNet [16]	41.5 ± 0.7	57.1 ± 0.8
MatchNet [17]	43.9 ± 0.7	57.1 ± 0.7
TADAM [43]	40.1 ± 0.4	56.1 ± 0.4
DeepEMD [31]	45.4 ± 0.4	61.5 ± 0.6
IDeMe-Net [44]	46.2 ± 0.8	64.1 ± 0.4
MixtFSL [45]	44.9 ± 0.6	60.7 ± 0.7
IER [46]	47.4 ± 0.8	64.4 ± 0.8
MCRNet [47]	41.0 ± 0.6	57.8 ± 0.6
MTL [48]	46.1 ± 0.8	61.4 ± 0.8
GFF (ours)	**49.2 ± 0.8**	**67.5 ± 0.8**

**Table 5 sensors-22-07640-t005:** Classification accuracies (%) with different methods in the 5-way K-shot cases on CUB and DOGS.

Methods	CUB		Dog	
	5-Way 1-Shot	5-Way 5-Shot	5-Way 1-Shot	5-Way 5-Shot
MatchNet [8]	57.6 ± 0.7	70.6 ± 0.6	45.0 ± 0.7	60.6 ± 0.6
ProtoNet [7]	53.9 ± 0.7	70.8 ± 0.6	42.6 ± 0.6	59.5 ± 0.6
RelationNet [9]	58.9 ± 0.5	71.2 ± 0.4	43.3 ± 0.5	55.2 ± 0.4
MAML [4]	58.1 ± 0.4	71.5 ± 0.3	44.8 ± 0.3	58.6 ± 0.3
DN4 [33]	55.2 ± 0.9	74.9 ± 0.6	45.7 ± 0.8	**66.3 ± 0.7**
PABN [49]	61.1 ± 0.4	76.8 ± 0.2	45.6 ± 0.7	61.2 ± 0.6
CovaMNet [32]	58.5 ± 0.9	71.2 ± 0.8	49.1 ± 0.8	63.0 ± 0.6
LRPABN [50]	**63.6 ± 0.8**	76.1 ± 0.6	45.7 ± 0.8	60.9 ± 0.7
GFF (ours)	61.1 ± 0.4	**78.6 ± 0.3**	**49.8 ± 0.8**	65.3 ± 0.8

## Data Availability

This paper is accompanied by the source code of the proposed algorithm publicly available at: https://github.com/Zhou-xy99/GFF (accessed on 10 September 2022).

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
