# Peer review of "Few-Shot Fine-Grained Image Classification via GNN"

_sensors, 2022, doi:10.3390/s22197640_

Round 1

Reviewer 1 Report

The paper is interesting and well-written. The proposal is sound and description are exhaustive, including figures, algorithms and mathematical expressions. Besides, conclusions are supported by results and results are coherent. They show a clear improvement compared to the state of the art technologies. 

My only concers is the validity of the proposed solution. Maybe it is not as success in order scenarios and types of pictures or images. Then, I strongly recommend the authors to discuss the validity threats of the proposed technology and the associated experimental validation.

But, in general, I think the paper may be accepted

Reviewer 2 Report

Your paper should have some numerical or experimental illustrations. In particular, the numerical experiments must have scientific value of their own. Furthermore, experimental comparisons with other approaches are strongly encouraged.

Your paper could improve in a number of areas such as a more thorough discussion of the design, development, testing and evaluation results; clarifying the key significance of the research contribution; ascertaining that the research fits the aims and scope of the journal; and a better command and flow of English writing throughout the paper.  

The references should be updated with the most recent in your paper's research field of relevance. I recommend the authors to consult the following survey and empirical papers to contextualize your findings. This should help the readers to understand the novelty of your work. 

Classifying With Adaptive Hyper-Spheres: An Incremental Classifier Based on Competitive Learning, IEEE Transactions on Systems, Man, and Cybernetics: Systems, DOI: https://doi.org/10.1109/TSMC.2017.2761360

An efficient stock market prediction model using hybrid feature reduction method based on variational autoencoders and recursive feature elimination. Financ Innov 7, 28 (2021). https://doi.org/10.1186/s40854-021-00243-3

Analysing the behavioural finance impact of 'fake news' phenomena on financial markets: a representative agent model and empirical validation. Financ Innov 7, 53 (2021). https://doi.org/10.1186/s40854-021-00271-z

Recent innovation in benchmark rates (BMR): evidence from influential factors on Turkish Lira Overnight Reference Interest Rate with machine learning algorithms. Financ Innov 7, 44 (2021). https://doi.org/10.1186/s40854-021-00245-1

Reviewer 3 Report

Please solve the following comments with a major revision and to be reviewed again.

1. In P3, L48, the author should supplement more GNN-based classification algorithms.

2. Why not add related work of GNN in Sec. 2.

3. Fig. 2 provides too little information, please redraw it.

4. Increase the readability of Fig. 3. For example, where is the GNN model in Fig. 3?

5. From P6, L202-L205, one cannot distinguish the function difference between Adjacency Block and Convolution Block.

6. In Table 2, it is confusing to see the “Enhancement”, which requires the relevant explanation.

7. In P3, L48, “It is of great significance to study this problem.”, So, Does the author study this problem and draw any conclusions?

8. The author needs to clarify the baseline of GFF.

9. I wonder if the author can try to conduct few-shot remote sensing image classification via GNN.

10. It will be welcome the authors could discuss some works in the background e.g. a mask attention interaction and scale enhancement network for sar ship instance segmentation, htc+ for sar ship instance segmentation, a polarization fusion network with geometric feature embedding for sar ship classification, balance learning for ship detection from synthetic aperture radar remote sensing imagery and so on.

11. IMHO, the Conclusion should be re-written to 1) explicitly describe the essential features/advantages of the proposed method that other methods do not have, 2) describe the limitation(s) of proposed method, and 3) what aspect(s) of the proposed method could be further improved, why and how.

12. The English should be improved greatly.

Round 2

Reviewer 2 Report

I have looked at the revised submission. I appreciate author's efforts to present their revised research work nicely, and I am satisfied with the efforts they employed in the revision. The authors have addressed the main remarks and comments, following the Reviewers' suggestions, and including them appropriately in the new version of the paper. The subject is interesting and the final aim of the contribution is clear. As for the paper's content, its structure is correct; it is easy to read; it contains all the relevant and necessary information for the reader. Overall, I am satisfied with the authors reply and I think this new version of the paper has earned in terms of completeness and it looks more solid with respect to previous submission. It is my pleasure to recommend for acceptance without any more changes. 

Reviewer 3 Report

Fine. Accept. No more comments.